# Regulation of Cellular Ribonucleoprotein Granules: From Assembly to Degradation via Post-translational Modification

**DOI:** 10.3390/cells11132063

**Published:** 2022-06-29

**Authors:** Pureum Jeon, Hyun-Ji Ham, Semin Park, Jin-A Lee

**Affiliations:** Department of Biotechnology and Biological Sciences, Hannam University, Daejeon 34054, Korea; vnfma1222@gmail.com (P.J.); hhgjr0305@gmail.com (H.-J.H.); smpark488@gmail.com (S.P.)

**Keywords:** RNP granules, stress granules, P-bodies, Cajal bodies, paraspeckles, autophagy, post-translational modification, neurodegenerative disease

## Abstract

Cells possess membraneless ribonucleoprotein (RNP) granules, including stress granules, processing bodies, Cajal bodies, or paraspeckles, that play physiological or pathological roles. RNP granules contain RNA and numerous RNA-binding proteins, transiently formed through the liquid–liquid phase separation. The assembly or disassembly of numerous RNP granules is strongly controlled to maintain their homeostasis and perform their cellular functions properly. Normal RNA granules are reversibly assembled, whereas abnormal RNP granules accumulate and associate with various neurodegenerative diseases. This review summarizes current studies on the physiological or pathological roles of post-translational modifications of various cellular RNP granules and discusses the therapeutic methods in curing diseases related to abnormal RNP granules by autophagy.

## 1. Introduction

Cells comprise membrane-bound organelles, including mitochondria, the Golgi apparatus, and the endoplasmic reticulum, and membraneless organelles, such as stress granules (SGs), processing bodies (P-bodies), Cajal bodies, and paraspeckles [1]. Many membraneless organelles contain RNAs and RNA-binding proteins (RBPs) (Figure 1), which are transiently formed through the liquid–liquid phase separation (LLPS), similar to oil droplet formation in water. The LLPS occurs due to the forces that weaken the electrostatic, hydrophobic, protein–protein, protein–RNA, and RNA–RNA interactions [2,3]. Notably, intrinsically disordered regions (IDRs) on RBPs are the key drivers of LLPS [4,5]. Generally, post-translational modifications (PTMs) of proteins in the various RNA granules, via phosphorylation, glycosylation, or methylation, impact the associations of RNP granule proteins, enzymatic activities, the stability of modified proteins, or intracellular locations. Remarkably, current accruing evidence proposes that PTMs altering the phase separation of these RBPs might play a crucial role in forming pathological RBP inclusions in neurodegenerative diseases [6]. Additionally, autophagy, a lysosomal degradation pathway, controls ribonucleoprotein (RNP) granules and pathological RBP aggregates.

This review summarizes several RNP granules and discusses the pathophysiological roles of PTMs of RNP granules that autophagy can regulate.

## 2. RNP Granules

### 2.1. Cytosolic RNP Granules

#### 2.1.1. SGs

SGs are dynamic cytoplasmic RNP granules formed during cellular stress, such as heat shock, oxidative stress, viral infection, osmotic stress, ultraviolet irradiation, or long-term starvation [7]. The major function of SGs is protection from the RNA damage and abnormal protein synthesis from stress conditions and to increase cell viability [8].

Notably, these SGs rapidly disassemble when stress is relieved. They generally contain RNAs, RBPs, and 40S ribosomal subunits and comprise cell-type- or stress-specific components [9]; however, SGs strictly require translation machinery and they are disrupted by translation inhibitors [10].

Many signaling molecules tightly regulate the assembly or disassembly of SGs. SG formation is mostly dependent on eukaryotic translation initiation factor 2α (eIF2α) phosphorylation using protein kinase RNA (PKR), heme-regulated inhibitor, general control nonrepressed 2 (GCN2), or PKR-like endoplasmic reticulum kinase. Most stress increases eIF2α (S51) phosphorylation, a component of the eIF2-GTP-tRNAiMet ternary complex, leading to the inhibition of protein translation [11]. SG assembly is enhanced by mTORC1 activation, which is regulated by PI3K and p38 [12]. SG formation is also affected by the mTORC1 effector kinases S6K1 and S6K2 via eIF2α phosphorylation regulation and persistence, respectively [13,14]. However, eIF4A activity or cap-dependent ribosome recruitment inhibition induces SG formation in an eIF2α phosphorylation-independent manner [15]. Intriguingly, various proteins are essential for SG formation, such as G3BP1/2 [16], TIA-1 [17], histone deacetylase 6 (HDAC6) [18], and UBAP2L [19]. These proteins generally possess IDRs, which regulate LLPS [17,20,21]. Many mutations in the IDRs of RBPs, such as FUS, TDP-43, EWS, and hnRNPA1, can alter the phase separation of RNA-binding proteins, consequently inducing gel-like structures/aggregates.

SGs are biphasic structures with a stable core that is usually surrounded by a dynamic shell [22]. They initially assemble stable cores, providing a platform for the growth of dynamic shells around these cores. Shell dissipation occurs first, followed by core decomposition during the disassembly step [23]. SG disassembly occurs in a reverse process where a less stable shell initially dissolves, accompanied by core disassembly and/or clearance by autophagy. Several corollaries exist in this observation. First, mRNA is believed to be in rapid equilibrium between the cytosol and SGs upon translation re-establishment. This exchange influences stress granule structural integrity and may account for titrating select RNA into translation. The lag in the clearance of granule cores may reflect the requirement of a myriad of ATP-dependent remodeling complexes (e.g., heat shock protein 70 or p97/valosin-containing protein (VCP) AAA-ATPase complexes [24,25]) and serve as a cytoprotective mechanism to acutely re-nucleate SGs whenever the cell re-encounters stress [23].

#### 2.1.2. P-bodies

P-bodies are also cytosolic dynamic RNP granules containing translationally repressed mRNAs and proteins involved in mRNA decay, increasing the roles of P-bodies in post-transcriptional regulation. P-bodies are compositionally and functionally associated with SGs [26]. They share numerous proteins, such as eIF4E, XRN1, and RAP55, which are usually fused [27,28], although the physiological meaning of their docking remains controversial. Though P-bodies and SGs are involved in post-transcriptional regulation and translational control, they form through the coassembly of translationally inactive mRNAs linked with distinct RNA-binding proteins (RBPs), notably with translational initiation components (SGs) and mRNA degradation machinery components (P-bodies) [29]. Unlike SGs, they are internally present within cells, further increased by cellular stresses, such as sodium arsenite or mild cold shock [30]. It has been reported that mRNA decapping and 5′–3′ decay generally promote P-body formation [31]. However, the mechanistic explanations for P-body formation are still unclear.

P-bodies are involved in mRNA storage and degradation, nonsense-mediated mRNA decay, translational repression, or microRNA (miRNA)/small interfering RNA-mediated gene silencing [32,33]. Interestingly, P-bodies drive viral mRNP inclusions to prevent viral protein synthesis during viral infection conditions.

Several factors regulate P-body assembly and disassembly. Notably, the formation and integrity of P-bodies require mRNA [34]. These mRNAs dissociate from the ribosome via translation inhibition, subsequently increasing P-bodies [26], and the overexpression of non-translating mRNA fragments increases the number and size of P-bodies. The various mRNA-regulating proteins involved in exonuclease activity (XRN1), decapping (Dcp1A, Dcp2, and Edc3), and deadenylation (Ccr4, Caf1, Pan2, and Pan3) are linked with the regulation of P-bodies. Similarly, compared with other key factors, 4E-T [35,36], GW182 [37,38], and RAP55 [27,39] can reduce the P-body assembly. Current studies reveal that the RNA helicase and P-body component, DDX6, modulates P-body homeostasis and controls the self-renewal and differentiation of stem cells [40].

#### 2.1.3. Germ-Cell-specific RNP Granules

Germ-cell-specific RNP granules are cytoplasmic RNP granules specifically formed only in germ cells for proper germ cell development from worms to humans [41]. Germ granules have various names depending on the organisms, such as chromatoid bodies in mammals, P granules in *Caenorhabditis elegans*, cytoplasmic polar granules in *Drosophila melanogaster*, and Balbiani bodies in zebrafish. Germ-cell-specific RNP granules act as storage compartments that prevent maternally provided mRNAs’ premature translation in many organisms [42,43]. Notably, germ granules provide platforms for the PIWI-interacting RNA (piRNA) pathway and are involved in piRNA biogenesis and piRNA-targeted RNA degradation. Recently, other RNA regulatory mechanisms, such as the nonsense-mediated mRNA decay pathway, have been linked with germ granules. Germ-cell-specific RNP granules share components with SGs and P-bodies and contain specific proteins and RNAs involved in gametogenesis and embryonic development [44]. DEAD-box RNA helicase VASA (DDX4) and nanos mRNA are vital components of the general germ granule. Among germ-cell-specific RNA granules, chromatoid bodies are transiently and highly condensed structures at the initial spermatogenesis stages, comprising mouse VASA homolog (MVH), MIWI, TDRD family proteins, UPF2, IP6K1, GRTH [45], and small RNAs, such as miRNAs and piRNAs.

The molecular specifics of germ-cell-specific RNP granule assembly and disassembly are poorly elucidated. However, a few studies reported that PGL proteins, PGL-1 and PGL-2, are crucial for germ granule formation through self-association and RBP recruitment [46]. Furthermore, in *Drosophila* germ granules, also called polar granules, RNA molecules form homotypic clusters containing multiple molecules of the same RNA species, signifying the importance of RNA assembly in the polar granules [47].

### 2.2. Nuclear RNP Granules

#### 2.2.1. Cajal Bodies

Cajal bodies, also called coiled bodies, are subnuclear organelles with diameters between 0.3 and 1.0 μm in the nucleoplasm of animal and plant cells. These are the main sites for assembling small nuclear RNPs (snRNPs). Thus, Cajal bodies are enriched with snRNPs, small nucleolar RNPs (snoRNPs), and telomerase RNA. Most snoRNPs are involved in guiding RNA modification in snoRNAs, specifically named scaRNAs in Cajal bodies [48]. These RNP granules are essential for histones mRNA, rRNA processing, and intron-encoded snoRNP biogenesis.

Coilin phosphorylation regulates the assembly and disassembly of the Cajal body. Coilin (also known as P80C), a marker protein of Cajal bodies, is essential for the proper formation of Cajal bodies and cell proliferation [49,50]. However, coilin overexpression does not increase the number or size of Cajal bodies, indicating that coilin concentration is not a limiting factor for their assembly [51]. Another regulator of the formation of Cajal bodies is WRAP53, also a central player in the trafficking and formation of telomerase RNPs [52]. Spinal motor neuron (SMN) protein, through its interaction with coilin, is also localized to Cajal bodies [53]. This protein is required for the biogenesis of various snRNPs. Therefore, SMN deletion inhibits Cajal body formation and maintenance [54]. Furthermore, Cajal bodies facilitate telomerase recruitment to telomeres and telomere elongation [55,56,57]. In this process, coilin and WRAP53 are associated with telomerase RNA, which may contribute to telomerase biogenesis.

#### 2.2.2. Paraspeckles

Paraspeckles are unique subnuclear structures composed of specific long noncoding RNAs (lncRNAs) and proteins [58]. They are structured in a core–shell spherical structures with irregular diameters of ~0.36 µm on average [59]. Paraspeckles are critical for controlling gene expression as the site of nuclear retention of specific mRNAs during differentiation, viral infection, and stress responses [60]. Additionally, they are involved in transcriptional gene regulation, such as mRNA biogenesis, pre-mRNA 3′-end formation, and cyclic AMP signaling.

Paraspeckles have lncRNAs, and nuclear paraspeckle assembly transcript 1 (NEAT1), composed of two isoforms (NEAT1_1 and NEAT1_2), serves as an architectural component [61]. Indeed, most paraspeckles are assembled near the NEAT1 locus on human chromosome 11 [62]. NEAT1 knockout mice fail to form paraspeckles, and only NEAT1_2 overexpression can rescue the paraspeckle formation, indicating that NEAT1_2 is essential for paraspeckle formation [63]. Interestingly, a recent study showed that TDP-43 and FUS, associated with amyotrophic lateral sclerosis (ALS), are abundant in paraspeckles and interact directly with NEAT1_2 in ALS-patient-derived motor neurons [64].

Some of the paraspeckle proteins overlap with SGs. Furthermore, stress-induced SG assembly induces paraspeckle formation, indicating that SGs act as key regulators of paraspeckle assembly in response to stress [65]. However, the mechanism for the disassembly of paraspeckles remains unknown.

#### 2.2.3. Promyelocytic Leukemia (PML) Bodies

PML bodies are spherical structures in the nucleus that can measure up to 1 µM in diameter. Unlike other granules, PML bodies only contain more than 170 proteins and do not contain RNA or DNA processes [66]. However, in a specific situation, such as HSV-1 infection into cells, the HSV-1 genome can be encased into PML bodies [67]. PML bodies vary in composition and are implicated in cellular processes such as telomere lengthening, DNA repair, and the DNA damage response [68,69,70]. Proteasomes and nuclear diffused defective ribosomal products (DRiPs) can accumulate in PML bodies with polyubiquitin conjugation, indicating that they remove newly synthesized misfolded proteins [71]. Moreover, the interaction of PML bodies with chromatin is important for controlling specific cellular chromatin assembly pathways and the chromatinization of viral genomes [67].

The principal organizing component of PML bodies is the PML protein, a member of the TRIM/RBCC family protein and a type of tumor suppressor gene. They are essential for the biogenesis and formation of PML bodies. This protein contains an N-terminal RING finger, two zinc fingers, and an RBCC domain. The RBCC domain mediates protein interaction, multimerization, and localization in the PML bodies. However, it does not have a DNA-binding capacity [72]. In particular, it is dimerized and oligomerized upon oxidative stress. Additionally, PTMs on PML are critical for PML body assembly. SUMOylation (small ubiquitin-like modifier) can alter protein interactions and cellular localization. The SUMOylation of PML also plays a critical role in recruiting partner proteins and accelerates PML body formation [73]. The SUMOylation of PML K160 is required for PML body recruitment [74]. Moreover, other SUMOylated proteins, such as Sizn1, are implicated in the accumulation of PML bodies [75]. Furthermore, DNA damage or stress-activated kinases, such as ATM [76], CHK2 [70], CK2 [77], or ERK [78], phosphorylate PML, regulating PML stability, PML body biogenesis, and interacting protein association.

#### 2.2.4. Nuclear SG-like Structures

Nuclear SG-like structures have been recently identified. Nuclear SG-like structures are assembled upon heat shock or stress induced by chemicals. Tellurite-induced oxidative stress also induces eIF2α phosphorylation and G3BP1- and eIF2β-positive nuclear granules [79]. Therefore, we called them nuclear SG-like structures. Nuclear SG-like structures are composed of the transcription and splicing machinery, RBPs, and satellite III (SatIII) lncRNAs [80,81]. Additionally, pre-mRNA-splicing factor SF2, an alternative splicing factor (SF2/ASF), is recruited to nuclear SG-like structures, which is mediated by a direct interaction with SatIII transcripts [82]. However, the specific roles of nuclear SG-like structures remain elusive.

#### 2.2.5. Physiological Roles of RNP Granules

RNP granules are eukaryote-conserved biomolecular condensates composed of RNA and RBPs, which play a major role in RNA metabolism. Moreover, in the case of cytoplasmic RNP granules, such as SGs, P-bodies are enriched with mRNAs that are suggested to play a role in translation regulation and mRNA storage. In this section, we describe the physiological roles of RNP granules.

RNP granules form and play roles in various cellular functions. P-bodies regulate mRNA metabolism, Cajal bodies play a role in RNA processing and telomerase biogenesis, and paraspeckles control gene expression. Among the RNP granules, SGs, which are the most studied, are mainly formed to cope with abnormal translation and improve cell survival in stress conditions. [1]. SGs can sequester various intracellular components and regulate cell signaling pathways. Stress can typically induce apoptosis to avoid stress-induced alterations. However, some evidence shows that SGs prevent apoptosis. The apoptotic signaling receptor protein RACK1 is sequestered into SGs under heat stress conditions, inhibiting the p38 and c-Jun N-terminal kinase apoptotic signaling pathway [2]. SGs also inhibit apoptosis by recruiting mammalian TOR (mTOR) to SGs [3]. Not only can SGs prevent apoptosis but they can also play a potential protective role through anti-inflammatory [4] and antioxidant effects [5].

A growing body of evidence implies that RNP granules carry specific mRNAs to target sites to synthesize the required proteins. Notably, highly polarized cells carry specific mRNAs to target sites to synthesize the required proteins in the neurons. In neurons, highly polarized cells, the transport of mRNAs to distal sites from the cell body is required for the accurate synthesis of specific proteins [6]. RNP granules containing mRNAs have been transported in neurons by interacting with the cytoskeleton and motor proteins. Additionally, RNP granules play a role in local protein synthesis by transporting mRNA to the axons or dendrites [7]. For example, the persistence of activated spine localized β-actin mRNA [8] undergoes multiple translations, and the newly synthesized proteins may function in spine enlargement and synaptic consolidation [6]. Furthermore, upregulated TDP-43 forms RNP granules, also called myo-granules, in injured skeletal muscle, which contains mRNA encoding structural proteins vital for proper muscle formation [9].

Further studies on the composition, assembly, and transport of RNP granules are required to understand their role in cellular regulation and organization.

## 3. Dynamic Regulation of RNP Granules by PTMs

As described earlier, many different RNP granules are dynamically assembled or disassembled, which can be tightly controlled in various ways for their proper cellular functions. Therefore, among the numerous ways to dynamically control RNP granules, the physiological PTM of RNP granules becomes an important factor (Table 1). Furthermore, currently accumulating evidence suggests that abnormal PTMs of RNP granules are also present in a disease state, leading to an alteration in the dynamics of RNP granules (Figure 2 and Figure 3).

PTMs refers to the covalent modification of amino acid residues using methyl, acetyl, phosphoryl, etc. PTMs generally regulate biological processes by affecting the interaction strength between proteins and nucleic acids, affecting protein stability and localization. This section will describe PTMs on RBPs to regulate RNP granule dynamics.

### 3.1. PTMs in Dynamic Physiological Regulation

#### 3.1.1. Arginine Methylation

Methylation by arginine methyltransferases or histone lysine methyltransferases is modified by the methyl group of various amino acid side chains. The methylation of RGG/RG motifs in RBPs by arginine methyltransferases regulates the phase separation to form RNP granules [83,84]. The arginine methylation of RNA-binding proteins reduces the arginine-π aromatic interaction and phase separation [85]. For example, the most common SG component, G3BP1, is methylated by arginine methyltransferases PRMT1 and PRMT5 [86]. JMJD6 (Jumonji C domain-containing protein), arginine demethylase, is known to demethylate G3BP1 directly or indirectly [87]. Previous studies showed that G3BP1 demethylation resulted in enhanced SG assembly. UBAP2L, an essential component for SG assembly and disassembly, is asymmetrically dimethylated by PRMT1 in its RGG motif [19,84]. Increased arginine methylation of UBAP2L inhibits SG formation. Altogether, arginine methylated SG core proteins can inhibit SG assembly. The dynamics of SGs and P-bodies or chromatoid bodies are also regulated by methylation. In contrast to SGs, PRMT5-mediated symmetrical dimethylarginine on Lsm4 [88] or SMEDWI-3 [89] promotes the formation of P-bodies or chromatoid bodies, respectively.

#### 3.1.2. Phosphorylation

Phosphorylation transfers a phosphate group from ATP to the receptor residues by kinase enzymes. Phosphorylation modification mainly occurs on Ser, Thr, Tyr, and His residues. Many RBPs are modified by phosphorylation and affect the dynamics of RNP granules. As one of the critical translation regulators during stress, Gle1 [90] is phosphorylated by MAPKs and GSK3 [91]. Phosphorylated Gle1 inhibits SG assembly and promotes disassembly. ULK1/2 and CK2 phosphorylate VCP [92] and G3BP1 S147 [93], respectively, which can induce efficient SG disassembly. Furthermore, Grb7, which is required to form SGs, is phosphorylated by focal adhesion kinase and attenuates SG disassembly [94]. Phosphorylation also affects the P-body assembly. The phosphorylated P-body component Dcp1A at S315 increases the number of Dcp1A-positve P-bodies [95]. Tristetraproline (TTP), a common component in SGs and P-bodies, is also phosphorylated by MAP kinase-activated protein kinase 2 (MK2). Interestingly, TTP phosphorylation by MAPKAP2 deliberates the protein from SGs but not from PBs [96]. In the phosphorylation of the protein-associated splicing factor and p54nrb, paraspeckle proteins regulate the integrity and size of paraspeckles [97,98].

#### 3.1.3. Acetylation

Protein residues modify acetylation with the acetyl group to the lysine side chains, which is catalyzed via lysine acetyltransferase and histone acetyltransferase. In SG dynamics, the acetylation of the lysine 376 residue in the RNA-binding domain of G3BP1 is regulated by HDAC6 and CBP/p300. Acetylated G3BP1 impaired RNA-binding and interactions with other components, facilitating SG disassembly [99]. The acetylation of the IDR of DDX3X, an SG component, impairs SG formation [100]. Furthermore, MVH (also known as DDX4, a germ-cell-specific-type RBP) in the chromatoid body of germ cells is acetylated in a Hat1-dependent manner during mammalian spermatogenesis, which leads to the inhibition of MVH RNA-binding capacity [101]. Most of the acetylation seems to regulate RNP granules negatively. However, the acetylation of SG components, such as FUS or TDP-43, is also linked to a pathological transition and will be described later.

#### 3.1.4. Glycosylation or Poly(ADP-ribosyl)ation (PARylation)

Interestingly, glycosylation is also one of the PTMs for regulating RNP granule dynamics. Glycosylation is the enzymatic process that modifies specific residues with oligosaccharide chains. Based on the target residue, it is classified into six groups, including N-glycosylation, O-glycosylation, and C-glycosylation. O-GlcNAc glycosylation regulates SG and P-body formation [102]. For example, the glycosylation of RACK1 and prohibitin-2, which are associated with the translational machinery, promotes SG assembly [102]. Conversely, a knockdown of GFAT, which converts glucose to GlcNAc and O-GlcNAc transferase, inhibits SG assembly. As with other PTMs, PARylation is a reversible process that adds ADP-ribose to the Glu, Asp, Lys, Arg, or Ser residue. PARylation is also known as the major SG dynamic regulator. The inhibition of PARylated TDP-43 or hnRNPA1 reduces the dynamics of SG assembly and disassembly [103].

#### 3.1.5. Ubiquitination, Neddylation, or SUMOylation

Ubiquitination, the addition of monoubiquitin or a ubiquitin chain on a single lysine residue on the substrate protein, affects cellular processes by regulating protein degradation via the proteasome. Neddylation is mediated by NEDD8, a small ubiquitin-like protein covalently conjugated to the lysine residue on the target protein, to induce a conformational change and provide a novel binding surface [104]. These PTMs are potential regulators of SG dynamics. SGs colocalize with ubiquitin–proteasome system-related proteins, including ubiquitin-specific protease 10, HDAC6, or ubiquitin-conjugating enzyme 9. Kwon et al. showed that polyubiquitinated G3BP1 is required for SG disassembly upon heat shock [105]. For SG assembly, serum and arginine-rich splicing factor 3(SRSF3), which is neddylated upon oxidative stress, is required [106]. Moreover, current studies reveal that ubiquitination or neddylation inhibition regulates SG dynamics [106,107,108]. However, there is a controversial study that found that the inhibition of ubiquitin-activating enzyme or NEDD8-activating enzyme did not affect SG dynamics [109]. SUMO protein is covalently attached to a lysine residue. SUMOylation is also a similar process to ubiquitination and regulates RNP granule dynamics. The SUMOylation of the PML protein enhances PML body formation, recruiting proteins [73].

#### 3.1.6. Hypusine Modification

Hypusine is a polyamine-derived amino acid that is essential for eukaryotic translation [110]. This target protein modification is a conjugation of the aminobutyl moiety of spermidine to a specific lysine residue. The post-translational modification is catalyzed by deoxyhypusine synthase (DHPS) and deoxyhypusine hydroxylase (DOHH). eIF5A is required for global protein synthesis, and only hypusine modifies protein. Furthermore, one study showed that a hypusine modification of eIF5A is required for arsenite-induced stress granule assembly, showing that a knockdown of DHPS inhibits the assembly of SGs [111].

Further detailed studies are required to address how many PTMs affect the dynamics of various RNP granules in a context-dependent manner.

### 3.2. PTMs on Pathological RNP Granules

Interestingly, many PTMs, such as PARylation, phosphorylation, or ubiquitination, mainly modify RBPs in RNP granules. Liquid-like RNP granules can alter gel-like or solid structures/aggregates by some PTMs [113], such as phosphorylation, acetylation, and PARylation. Conversely, PTMs also inhibit RNP granules’ transition from liquid-like structures to aggregates. This section describes the regulation of pathological RNP granules by PTMs (Figure 3).

For example, the PARylation of TDP-43 and hnRNPA1 reduces SG dynamics, increases cytotoxicity, and promotes protein aggregation [103]. Tau is regulated by two PTMs that have the opposite effect on aggregation properties. The phosphorylation of tau promotes phase separation and can serve as an intermediate to aggregate formation [114]. However, the acetylation of tau decreases phase separation, preventing aggregate formation [115]. Furthermore, the methylation [85,116], phosphorylation [117,118,119], and acetylation [120,121] of FUS or TDP-43 associated with ALS and frontotemporal dementia (FTD) decrease aggregation. The phosphorylation of TDP-43 could induce or reduce aggregation in a context-dependent manner [122]. Intriguingly, the O-GlcNAcylation of TDP-43 inhibits its hyperphosphorylation, resulting in the suppression of TDP-43 aggregates [123]. Additionally, phosphorylation can induce the ubiquitination of RBPs within aggregates and promote proteasomal or autophagic degradation, showing that PTMs can promote other PTMs.

There are some RNP granules called myo-granules (again, TDP-43), transport granules, and neuronal granules in neurons. These mainly transport mRNAs to target sites and are important for neuronal functions such as axonal translation, synaptic properties, and memory. A very small portion of TDP-43 localizes to cytoplasmic RNA granules. TDP-43-positive RNA granules in neurons deliver the target mRNA to a distal neuronal compartment. TDP-43 mutations inhibit mRNA-RNA granule transport kinetics in Drosophila motor neurons, mouse cortical neurons, and ALS patient iPSC-derived neurons [124]. Interestingly, the retrograde movement was significantly decreased. This paper suggested that impairing the axonal transport of mRNA targeted by TDP-43 may cause ALS or FTD pathogenesis. PTM also regulates RNP granules in neurons and their function. Fragile X mental retardation protein (FMRP) binds mRNAs and generates RNP granules. These transport mRNA to active synapses along axons and dendrites to regulate local translation [125]. During this process, FMRP SUMOylation is essential for maintaining RNP granule shape and regulating spine elimination and maturation. Additional research is needed to uncover the therapeutic target for inhibiting the aggregate property, which is related to neurodegenerative diseases.

**Figure 3 cells-11-02063-f003:**
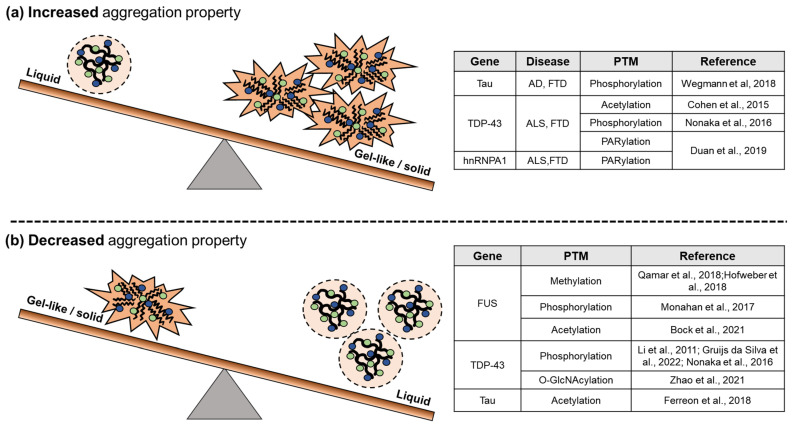
PTMs on pathological RNP granules. (**a**) Increased aggregation property by PTMs on RBPs: phosphorylation of tau [114] or TDP-43 [121], acetylation of TDP-43 [122], PARylation of TDP-43, or hnRNPA1 [103] accelerate the solid transition. (**b**) Decreased aggregation property by PTMs on RBPs: methylation [85,116], phosphorylation [119], or acetylation [120] of FUS; phosphorylation [117,118,122], or acetylation of tau [115] prevent the phase to solid transition.

## 4. Abnormal RNP Granules and Neurodegenerative Diseases

Ischemia, neuroinflammation, and aging affect SG formation and induce the irreversible accumulation of RBPs in SGs in neurons [126,127,128]. SGs are mainly composed of RBPs, which are proteins that frequently contain low-complexity domains. These domains make RBPs more prone to aggregate, interact with other proteins, and recruit to aggregates. Inside cells, aggregate-prone RBPs normally participate in the repeating and highly dynamic cycles of functional assembly and disassembly of protein-RNA granules such as SGs [25]. On the other hand, the pathological SGs cause RBPs and RNAs/DNAs to aggregate abnormally and stably. Their presence may disrupt the normal RBP aggregation equilibrium; RBPs from SGs and abnormal disease-containing proteins with low-complexity domains may produce an overactive aggregation phenomenon within neurons. The sequestration of RBPs and other components of SGs may contribute to the pathogenesis of several neurodegenerative diseases with aggregate-prone proteins. Cell stress and stress response dysfunction may further exacerbate abnormal protein disruption, further feeding the aggregation cascade. This section will describe the accumulation of abnormal RNP aggregates, a hallmark of several neurodegenerative diseases.

Current perspectives on the biology of proteins with low-complexity sequences suggest a putative mechanistic link between the biophysical properties of SGs and the pathologic aggregation of disease-associated proteins such as FUS, TDP-43, or poly-Q diseases. Proteins enriched in low-complexity sequences are prone to forming intracellular liquid-like condensates, of which SGs are an example [129]. Phase-separated membraneless organelles appear to be metastable. While they can disassemble, they can also evolve into a solid-like state or directly drive the formation of amyloid-like fibrils, as demonstrated for abnormal SGs [4]. Many RBPs associated with ALS or FTD, such as TDP-43 [130], FUS [131], Ataxin2 [132], C9orf72 [133], hnRNPA1 [134], and TIA-1 [135], are colocalized to SGs. Interestingly, several genetic mutations in the IDRs of RBPs result in the accumulation of abnormal SGs. For example, mutations on TDP-43 (A315T and M337V) [136], FUS (G156E) [137,138], TIA-1 (P362L) [135], and hnRNPA2B1 (hnRNPA2 D290V and hnRNPA1 D262V) [134] impair SG dynamics, leading to the accumulation of abnormal SGs. Additionally, poly(GR) dipeptide repeats in ALS-linked C9orf72 induce SG formation and delay disassembly [133]. The connection of SGs with neurodegeneration is not limited to changes in protein aggregates. The cytoplasmic mislocalization of TDP-43, which normally has a nuclear localization, is sequestered into SGs. Nucleocytoplasmic transport factors sequester into SGs and impair nucleocytoplasmic transport [139]. Furthermore, SGs that are spontaneously assembled by mutant TDP-43 and dipeptide repeat proteins also disrupt nucleocytoplasmic transport, contributing to the pathogenesis of ALS/FTD. Parkinson’s disease (PD)-associated protein, DJ-1 mutation, promotes abnormal SG formation and PD pathogenesis [140]. Furthermore, G3BP1-positive SGs that recruit TDP-43 are increased in huntingtin mutant-expressing Huntington’s disease (HD) mouse cortical neurons [141]. This implies that the abnormal dynamics of SGs play a significant role in the pathophysiology of HD, and mutations in RBPs cause persistent stress in cells.

Tau protein is a risk factor for neurodegenerative diseases, including Alzheimer’s disease. It is associated with microtubules and is essential for transport along axons. Although tau protein is not an RBP, they are localized to SGs, interacting with TIA1. Tau can promote SG formation, and interactions with TIA1 contributes to the pathophysiology of tauopathy [142]. Furthermore, tau phosphorylation can accelerate phase separation and tau aggregates [114].

Acute stress could also enhance the dynamics of SGs to support the physiological cellular environment. However, the SGs induced by chronic/prolonged stresses differ from acute-stress-induced SGs in their contents and dynamics [143]. For example, SGs induced by acute stress are translation initiation factors, 40S ribosomes, RBPs, and mRNAs, whereas SGs formed under chronic nutrient starvation lack 40S ribosomes and have reduced dynamics [144]. In addition, chronic arsenic stress also formed pathological TDP-43 aggregates in induced pluripotent stem cell (iPSC)-derived motor neurons from patients with ALS [145].

The accumulation of paraspeckles and neural RNA transport granules is associated with neurodegenerative diseases. Paraspeckles, such as SGs, also contain ALS-associated proteins, including TDP-43 and FUS. The sequestration of TDP-43 into RNP granules enhances paraspeckle assembly by the upregulation of NEAT1 and NEAT1_2 and the impairment of miRNA processing during ALS pathogenesis [146]. TDP-43 D169G mutation impairs the assembly of TDP-43-positive paraspeckles, causing an excessive translocation of TDP-43 into cytoplasmic SGs [147]. Additionally, FUS P525L mutation promotes the excessive formation of dysfunctional paraspeckles [148]. On the contrary, FUS R522G mutation induces the accumulation of cytoplasmic FUS-positive paraspeckles in motor neurons of patients with ALS [149]. The alteration of the dynamics of RNP granules by IDR mutations or chronic stresses induces persistent RNA granules resistant to degradation. The excessive accumulation of abnormal RNP granules might cause neuronal toxicity. Therefore, manipulating abnormal RNP granules is an excellent matter of therapeutic approaches. The following section will describe the removal of abnormal aggregates of RNP granules.

## 5. Autophagic Degradation of RNP Granules: Granulophagy

Current accruing evidence proposes that the disassembly and removal of RNP granules can be induced by chaperones, such as HSP70 family proteins, or autophagy. In particular, gel-like or solid RNP granules are resistant to disassembly. Therefore, intracellular degradation pathways should clear these abnormal structures. Among the degradation pathways, there is autophagy, a self-degradation pathway in which cytoplasmic materials are sequestered into double-membrane-bound vesicles and delivered to the lysosomes for degradation. RNP granules could be degraded by autophagy, which is called granulophagy [24,150,151].

Until now, there are a few proteins involved in granulophagy. For example, VCP belongs to the AAA+ (ATPases associated with diverse cellular activities) family of a chaperone-like protein involved in various cellular processes [152]. VCP promotes autophagosome biogenesis and is critical for SG clearance and interacting with LC3 and p62 [24,153,154]. Furthermore, hypoxia-induced SG-like structures in *Arabidopsis thaliana* [155] contain calmodulin-like 38, a calcium-sensor protein, and cell division cycle 48A (ortholog VCP in mammals). These proteins regulate their degradation by autophagy. More recently, ULK1 and ULK2, autophagy-inducing kinases, localize to SGs and regulate their disassembly by VCP phosphorylation [92]. This process is dependent on the enhanced ATP hydrolase activity of VCP. Moreover, VCP mutations are identified in ALS and FTD, suggesting that defects in VCP cause the abnormal accumulation of SGs due to the impairment of autophagy, leading to neurodegeneration [156,157].

Zinc finger AN1-type containing 1 (ZFAND1) is known to recruit 26S proteasome and VCP into arsenite-induced SGs for the clearance of SGs [158]. DRiP-positive abnormal SGs accumulate in ZFAND1-deficient cells treated with autophagy inhibitor, indicating that abnormal SGs can be disassembled or cleared by the proteasome and then activated autophagy to control protein homeostasis. Furthermore, a recent study showed that chronic oxidative stress increased the number of p62-TIAR- and LC3-TIAR-positive autophagosomes in TDP-43 A382T patient-derived iPSC motor neurons, suggesting that SG clearance is regulated by autophagy [145].

Autophagy can degrade P-bodies in yeast and mammalian cells [24,159]. The deletion of atg15 in yeast causes the accumulation of P-bodies [24]. An autophagy inhibitor blocked the transmission of growth factor-β (TGF-β)-induced Dcp1A-positive P-body clearance upon TGF-β removal conditions [159]. Another RNP granule, germ granules in *C*. *elegans*, termed PGL granule component PGL-3, interacts with SEPA-1, forming aggregates and binding to LGG-1/Atg8 [160,161].

Autophagy adaptor protein, p62 or NDP52, is preferentially colocalized with SGs or P-bodies [162]. Lee et al. [163] also showed that p62 knockout mouse embryonic fibroblasts slowly recover the SG from heat shock stress. Additionally, an autophagy inhibitor, bafilomycin A1, inhibits the clearance of heat-shock-induced SGs. p62, as an adaptor protein for SG degradation, is also required to degrade arsenite-induced SGs [150]. One of the autophagy genes, Atg19, is known to mediate cargo selection in selective autophagy [164]. SG and P-body proteins also colocalize to Atg19 [24]. Taken together, Atg19 may contribute to granulophagy by targeting RNP granules.

As mentioned earlier, the protein quality control system through granulophagy is important for the clearance of abnormal RNP granules. However, a chaperone-mediated protein quality system and granulophagy could prevent the formation of abnormal SGs and promote their disassembly [165].

DRiP (prematurely terminated and misfolded polypeptides) accumulation in SGs leads to aberrant SGs and delays their disassembly. A recent study also showed that SGs are regulated by a chaperone complex, HSPB9-BAG3-HSP70 [166]. However, the relative contribution of granulophagy or chaperone on the clearance of physiological or pathological RNP granules needs to be further studied.

## 6. Discussion

Many nuclear or cytoplasmic RNP granules, such as dynamic membraneless organelles, regulate protein synthesis or various RNA processes, which are important for regulating gene expression. Thus, the assembly and disassembly of these RNP granules must be tightly regulated by multiple cellular mechanisms. PTMs act as important regulators of RBPs and influence the biophysical properties, molecular interactions, subcellular localizations, and cellular functions of RBDs in RNP granules. This review has described the roles of physiological or pathological PTMs on various RNP granules. Accumulating evidence suggests that PTMs are emerging as crucial factors for the dynamics and assembly/disassembly of RNP granules in a context/tissue-dependent manner. Notably, further detailed proteomic studies might explain why components of RNP granules are different or how PTMs have a tissue-specific pattern in a physiological or pathological state. Moreover, how the combinational effects of PTMs within one protein affect the function of RNP granules will explain the PTM perspective in the regulation of RNP granules. Beyond the PTMs in RBPs discussed in this review, many other key regulators of PTMs might be associated with RNP granules, which will be an exciting future area of exploration for scientists to discover.

Additionally, this review also described granulophagy, which can regulate RNP granule clearance, including SGs. Many exciting questions must be solved to understand the cellular mechanism of granulophagy. For example, what are the specific signals for activating granulophagy for physiological or pathological RNP granules? Which PTMs could contribute to granulophagy? How effectively would a combinational approach that controls autophagy and PTMs treat neurodegenerative diseases, such as FTD and ALS? In addition to the regulation of cytosolic RNP granules, recent studies suggest the importance of nuclear autophagy for nuclear RNP granules. Future studies to uncover the molecular mechanism of nuclear RNP granules will open new fields for RNA metabolism, granulophagy, and neurodegenerative diseases.

## Figures and Tables

**Figure 1 cells-11-02063-f001:**
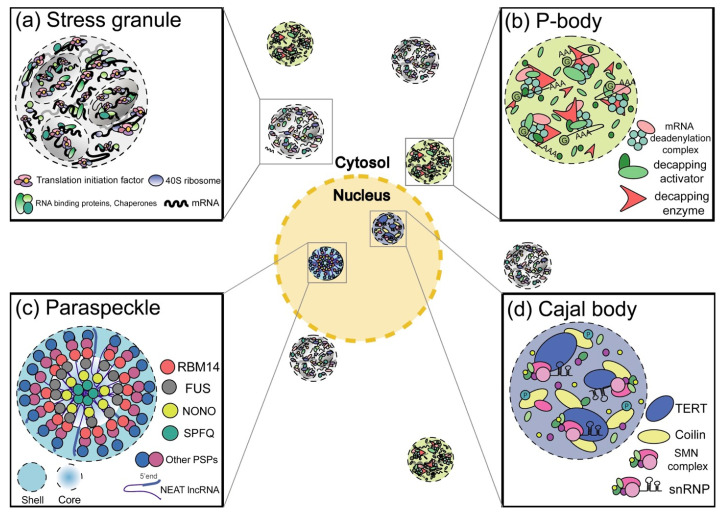
Cellular ribonucleoprotein (RNP) granules. (**a**,**b**) Schematic structural representation of cytoplasmic RNP granules. (**a**) Stress granules (SGs) containing untranslated mRNA, ribosomes, translational initiation factors, and RBPs. (**b**) P-bodies containing untranslated mRNAs and RBPs. (c and d) Schematic structure of nuclear RNP granules. (**c**) Paraspeckles containing lncRNA NEAT1 and nuclear-localized RBPs. (**d**) Cajal bodies containing snRNPs, snoRNPs, and nuclear RNPs.

**Figure 2 cells-11-02063-f002:**
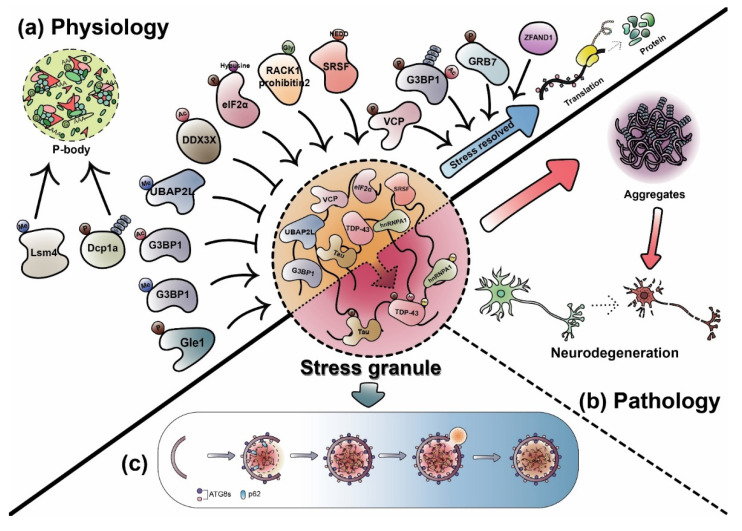
Physiological/pathological RNP granules regulated by PTMs. (**a**) Physiological RNP granules composed of proteins modified by PTMs, such as arginine methylation, phosphorylation, acetylation, ubiquitination, and glycosylation. PTMs can regulate RNP granule dynamics by affecting the interaction strength between proteins and nucleic acids. (**b**) RNP-binding proteins linked to neurodegenerative diseases can be modified by phosphorylation, acetylation, or PARylation (Figure 3), altering the biophysical properties. Accumulation of aggregates associated with altered RNP granules in neurons is a hallmark of several neurodegenerative diseases. (**c**) Abnormal RNP granules and aggregates can be degraded by granulophagy.

**Table 1 cells-11-02063-t001:** PTMs regulating physiological ribonucleoprotein (RNP) granule dynamics.

PTM	Protein	RNP Granule	Assembly or Disassembly	Reference
Arginine methylation	G3BP1	SG	Disassembly	[86,87]
	UBAP2L		Disassembly	[19,84]
	Lsm4	P-body	Assembly	[88]
	SMEDWI-3	Chromatoid body	Assembly	[89]
Phosphorylation	VCP	SG	Disassembly	[92]
	G3BP1		Disassembly	[93]
	Gle1		Assembly	[91]
	eIF2α		Assembly	[11]
	GRB7		Disassembly	[94]
	PSF, p54nrb1	Paraspeckle	Assembly	[97,98]
Acetylation	G3BP1	SG	Disassembly	[99]
	DDX3X		Disassembly	[100]
PARylation	TDP-43, hnRNPA1	SG	Delayed assembly and disassembly (low dynamics)	[103]
Phosphorylation and ubiquitination	Dcp1A	P-body	Assembly	[95]
Ubiquitination	G3BP1	SG	Disassembly	[105]
Neddylation	SRSF3	SG	Assembly	[106]
O-GlcNAcylation	RACK1, prohibitin-2	SG	Assembly	[102]
SUMOylation	PML	PML body	Assembly	[73,112]
Hypusine modification	eIF5A	SG	Assembly	[111]

## Data Availability

Not applicable.

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
