# Peer review of "Regulation of Cellular Ribonucleoprotein Granules: From Assembly to Degradation via Post-translational Modification"

_cells, 2022, doi:10.3390/cells11132063_

Round 1

Reviewer 1 Report

The review is timely and contains useful information about PTM regulations of RNP granules.

Below are several suggestions:

  • Overall, the role of PTMs and their types and regulations are well discussed. With that said, I feel that the primary physiological functions of the RNP granules themselves are still under debate in the fields and the review doesn’t much provide new insights on that matter.
  • As there are many similarities between SGs and P bodies, the review can provide more systematic “compare and contrast” between these granules.
  • Page 3 Cajal body: what are the functional links among snoRNPs, telomere, and coilin?
  • Is there a different name for “nuclear SG-like structures”? Does it refer to nuclear speckle or splicing speckle?
  • Page 4, lane 165: If PML body interacts with chromatin, why does PML not contain any RNA or DNA?
  • The intro on PML does not provide the functions of PML bodies. (if any, they are too brief)
  • Page5 Arginine methylation: what is the role of this modification (how does it regulate the assembly/disassembly of SGs)
  • For the PTM part, I feel that it is easier to follow if they are organized granule-by-granule, rather than by modification types.
  • Page 7, the role of SUMO on PML bodies is described too briefly.
  • Figure 2, it may be better to label that the tables indicate pathological modifications. Also indicate what types of diseases accrue by those modifications.
  • Figure 3: it may be better if the figure contains regulators of clearance, such as ZFAND1 or p62?
  • Provide citations: page2 (“the major functions of SGs it to protect RNA damage……”)

Author Response

The review is timely and contains useful information about PTM regulations of RNP granules.

Below are several suggestions:

  • Overall, the role of PTMs and their types and regulations are well discussed. With that said, I feel that the primary physiological functions of the RNP granules themselves are still under debate in the fields and the review doesn’t much provide new insights on that matter.

Response: Thank you for your comments. We added “2.2.5. Physiological roles of RNP granules” in the text of the revised manuscript (220-252 line, -5-6 page).

  • As there are many similarities between SGs and P bodies, the review can provide more systematic “compare and contrast” between these granules.

Response: Thank you for your suggestion. We added more detailed description for their differences in the revised manuscript.  (83-90 Line, 2 page).

  • Page 3 Cajal body: what are the functional links among snoRNPs, telomere, and coilin?

Response: Thank you for pointing out this matter. We added their functional link in the revised manuscript (154-156 line , 4 page)

  • Is there a different name for “nuclear SG-like structures”? Does it refer to nuclear speckle or splicing speckle?

Response: Thank you for your comment. Nuclear SG-like structures typically different from nuclear speckles or splicing speckles. We emphasized definition of nuclear SG-like structures in the revised manuscript (213 line, 5 page)

  • Page 4, lane 165: If PML body interacts with chromatin, why does PML not contain any RNA or DNA?

Response: Thank you for your question. We additionally mentioned PML bodies composition in normal and viral infection conditions in the revised manuscript (184-185 line, 4 page).

  • The intro on PML does not provide the functions of PML bodies. (if any, they are too brief)

Response: Thank you for your comment. We added more introduction for PML proteins in PML bodies in the manuscript (194-198 Line, 5 page).

  • Page5 Arginine methylation: what is the role of this modification (how does it regulate the assembly/disassembly of SGs)

Response: Thank you for your comment. We added the role of arginine methylation on assembly/disassembly of SGs in the text of the revised manuscript (282-283 line, 7 page).

  • For the PTM part, I feel that it is easier to follow if they are organized granule-by-granule, rather than by modification types.

Response: Thank you for your suggestion. However, in our manuscript, we wanted to focus more on post-translation modifications of various granules.

  • Page 7, the role of SUMO on PML bodies is described too briefly.

Response: Thank you for the comments. According to the reviewer’s suggestion, we added general role of SUMO conjugation on target protein and their role in PML bodies in the text of the revised manuscript (199-204 Line, 5 page).

  • Figure 2, it may be better to label that the tables indicate pathological modifications. Also indicate what types of diseases accrue by those modifications.

Response: Thank you for your suggestion. The table above shows pathological modifications and the table below shows a decrease in the cohesive properties for pathological forms by PTM. In addition, disease types have been added to the table in the revised manuscript (Revised Figure 2).

  • Figure 3: it may be better if the figure contains regulators of clearance, such as ZFAND1 or p62?

Response: Thank you for your suggestion. We added p62 and ZFAND1 as RNP granule regulator into revised Figure 3.

  • Provide citations: page2 (“the major functions of SGs it to protect RNA damage……”)

Response: Thank you for your comment. We added citation and revised the word ‘protect’ to ‘prevent’.

Reviewer 2 Report

The reviewer finds the manuscript potentially useful as an entry-level guide to the RNA-protein granules, but feels that a) the topic has not been covered sufficiently enough to make the manuscript stand out of other similar reviews, and b) certain basic things require (more) explanations.

To make the manuscript suitable for publications, I would kindly suggest to:

  1. Structure the text so that it becomes a narrative and not just a bundle of facts. It was not easy to wade through the manuscript.
  2. Discuss what is “abnormal SG formation”.
  3. Discuss how abnormal (pathological) granules differ from normal ones.
  4. Discuss how normal granules are disassembled.
  5. Add a brief description of myo-granules (again, TDP-43), transport granules and neuronal granules (especially as the part of the review is devoted to neurodegenerative diseases and PTM). Inhibiting PTMs of certain neuronal granules-associated proteins have been shown to affect axonal translation, synaptic properties and even memory, so ignoring this is unreasonable.

With respect to the list of PTMs, one rare yet important modification has not been mentioned. The translation factor eIF5A is modified by hypusine, and either eIF5A or the polyamine synthesis pathway genes KDs inhibit arsenic-induced SG formation (Ohn, 2008; Li, 2010). Definitely worth mentioning.

One distinctive feature of SGs has not been highlighted, while I feel it should be. That is, SGs strictly require ongoing translation for their assembly and translation inhibitors disrupt SGs. A prolonged emetine treatment also disassembles PBs. This shows that both these entities are highly dynamic. A reader may overlook this and have an impression that SGs are stable structures that assemble upon a stress and disassemble when the stress is gone. However, while individual SGs may exist for an hour or even longer, an average time of RNA or protein residence in SGs does not exceed minutes and generally is within the order of seconds. Also, the SG chapter may cause the impression that translation inhibition and SG formation are strictly linked processes, which is definitively not the case. Probably the most instructive here was the study of cold-shock SGs (Hofmann, 2012). It also should be noted that certain mRNAs evade incorporation into SGs (HSP70, SHP90). Likewise, it is probably worth mentioning that there are distinct subtypes of SGs, e.g., canonical and generally cytoprotective, lacking eIF2, and phospho-eIF2alpha-independent ones contain eIF2 and may be cytotoxic.

It is also not crystal clear from the text that SGs and the other granules share common components. As the manuscript describes the PTMs, it is interesting to note that TTP phosphorylation by MAPKAP2 deliberates the protein from SGs, but not from PBs.

When talking about the connection between the granules and neurodegeneration, a huge layer of information has been ignored. What are the corresponding mechanisms? We know that certain mutations in FUS or TDP-43 (predominately nuclear proteins) result in their mis-localization to cytoplasm, which in turn promotes their multimerization and granules formation. As some nucleocytoplasmic proteins tend to localize in SGs, formation of the latter affects nuclear transport (see C9orf72-mediated pathogeny). We know that granules formation can result in mRNA mislocalization in neurons. So, it’s not only the increased/decreased aggregation properties’ alterations.

Tau protein in associated with microtubules and is essential for transport along axons and its function and (mis)localization are indeed regulated by phosphorylation. But what’s a link between tau and granules? And, in first instance, what tau is? Such things should be explicitly described for a broader reader.

Minor comments

Lane 46           it should be pointed that only 40S, but not 60S ribosomal subunits are present in SGs.

Lanes 50-52    Please double check the sentence that starts with “The major function”.

Lane 60           eIF2a should be eIF2⍺ (the same applies to the Table)

Lane 230         RBPRBPs should be RBPs

Quite often a space is missed before brackets.

Ohn, T., Kedersha, N., Hickman, T., Tisdale, S. & Anderson, P. A functional RNAi screen links O-GlcNAc modification of ribosomal proteins to stress granule and processing body assembly. Nat. Cell. Biol. 10, 1224–1231 (2008).

Li, C. H., Ohn, T., Ivanov, P., Tisdale, S. & Anderson, P. eIF5A Promotes Translation Elongation, Polysome Disassembly and Stress Granule Assembly. Plos One 5, e9942 (2010).

Hofmann, S., Cherkasova, V., Bankhead, P., Bukau, B. & Stoecklin, G. Translation suppression promotes stress granule formation and cell survival in response to cold shock. Mol Biol Cell 23, 3786–3800 (2012).

Khayachi, A. et al. Sumoylation regulates FMRP-mediated dendritic spine elimination and maturation. Nat Commun 9, 757 (2018).

Qamar, S. et al. FUS Phase Separation Is Modulated by a Molecular Chaperone and Methylation of Arginine Cation-πInteractions. Cell 173, 720-734.e15 (2018).

Andrusiak, M. G. et al. Inhibition of Axon Regeneration by Liquid-like TIAR-2 Granules. Neuron 104, 290-304.e8 (2019).

Ford, L., Ling, E., Kandel, E. R. & Fioriti, L. CPEB3 inhibits translation of mRNA targets by localizing them to P bodies. Proc National Acad Sci 116, 18078–18087 (2019).

Author Response

The reviewer finds the manuscript potentially useful as an entry-level guide to the RNA-protein granules, but feels that a) the topic has not been covered sufficiently enough to make the manuscript stand out of other similar reviews, and b) certain basic things require (more) explanations.

To make the manuscript suitable for publications, I would kindly suggest to:

  • Structure the text so that it becomes a narrative and not just a bundle of facts. It was not easy to wade through the manuscript.

Response: Thank you for your constructive suggestion. According to the reviewer’s suggestion, we revised the text accordingly throughout the manuscript.

  • Discuss what is “abnormal SG formation”.

Response: According to the reviewer’s suggestion, we have discussed the process of abnormal SGs formation in the revised manuscript (425-432 Line, 10-11 Page).

  • Discuss how abnormal (pathological) granules differ from normal ones.

Response: According to the reviewer’s suggestion, we included discussion how pathological granules differ from normal ones (410-422 Line, 10 Page).

  • Discuss how normal granules are disassembled.

Response: According to the reviewer’s suggestion, we added the disassembly of normal SGs in the text of the revised manuscript (69-78 Line, 2 Page).

  • Add a brief description of myo-granules (again, TDP-43), transport granules and neuronal granules (especially as the part of the review is devoted to neurodegenerative diseases and PTM). Inhibiting PTMs of certain neuronal granules-associated proteins have been shown to affect axonal translation, synaptic properties and even memory, so ignoring this is unreasonable.

Response: Thank you for your suggestion. According to the reviewer’s suggestion, we added a a brief description of myo-granules (again, TDP-43), transport granules and neuronal granules. (392-406 Line , 10 Page)

  • With respect to the list of PTMs, one rare yet important modification has not been mentioned. The translation factor eIF5A is modified by hypusine, and either eIF5A or the polyamine synthesis pathway genes KDs inhibit arsenic-induced SG formation (Ohn, 2008; Li, 2010). Definitely worth mentioning.

Response: Thank you for constructive comments. According to the reviewer’s suggestion, we added ‘3.1.6. Hypusine modification’ section in the revised manuscript (359-369 Line, 8-9 Page)

  • One distinctive feature of SGs has not been highlighted, while I feel it should be. That is, SGs strictly require ongoing translation for their assembly and translation inhibitors disrupt SGs. A prolonged emetine treatment also disassembles PBs. This shows that both these entities are highly dynamic. A reader may overlook this and have an impression that SGs are stable structures that assemble upon a stress and disassemble when the stress is gone. However, while individual SGs may exist for an hour or even longer, an average time of RNA or protein residence in SGs does not exceed minutes and generally is within the order of seconds. Also, the SG chapter may cause the impression that translation inhibition and SG formation are strictly linked processes, which is definitively not the case. Probably the most instructive here was the study of cold-shock SGs (Hofmann, 2012). It also should be noted that certain mRNAs evade incorporation into SGs (HSP70, HSP90). Likewise, it is probably worth mentioning that there are distinct subtypes of SGs, e.g., canonical and generally cytoprotective, lacking eIF2, and phospho-eIF2alpha-independent ones contain eIF2 and may be cytotoxic.

Response: Thank you for your constructive suggestions. According to the reviewer’s comments, we added more detailed description regarding SGs. (49-50 Line, 2 Page)

  • It is also not crystal clear from the text that SGs and the other granules share common components. As the manuscript describes the PTMs, it is interesting to note that TTP phosphorylation by MAPKAP2 deliberates the protein from SGs, but not from PBs.

Response: Thank you for your comment. Following your kind information, we added description of TTP phosphorylation in phosphorylation section (307-309 Line, 7 Page).

  • When talking about the connection between the granules and neurodegeneration, a huge layer of information has been ignored. What are the corresponding mechanisms? We know that certain mutations in FUS or TDP-43 (predominately nuclear proteins) result in their mis-localization to cytoplasm, which in turn promotes their multimerization and granules formation. As some nucleocytoplasmic proteins tend to localize in SGs, formation of the latter affects nuclear transport (see C9orf72-mediated pathogeny). We know that granules formation can result in mRNA mislocalization in neurons. So, it’s not only the increased/decreased aggregation properties’ alterations.

Response: Thank you for pointing out this matter. We added this description into 4. Abnormal RNP granules and neurodegenerative disease section in the revised manuscript (439-445 Line, 11 Page).

  • Tau protein in associated with microtubules and is essential for transport along axons and its function and (mis)localization are indeed regulated by phosphorylation. But what’s a link between tau and granules? And, in first instance, what tau is? Such things should be explicitly described for a broader reader.

Response: We appreciate your comment. We described more information of Tau in the text of the revised manuscript (451-456 Line, 11 Page).

Minor comments

  • Lane 46           it should be pointed that only 40S, but not 60S ribosomal subunits are present in SGs.

Response: Thank you for your comment. We corrected this mistake in the revised manuscript (48 Line, 2 page)

  • Lanes 50-52    Please double check the sentence that starts with “The major function”.

Response: We appreciate this comment. We revised the sentence (to protect -> protect from) and added a reference. (46 Line, 2 Page)

  • Lane 60           eIF2a should be eIF2⍺ (the same applies to the Table)

Response: Thank you for your comment. We changed eIF2a to eIF2α.

  • Lane 230         RBPRBPs should be RBPs

Response: Thank you for the comment. RBPRBPs corrected to RBPs.

  • Quite often a space is missed before brackets.

Response: Thank you for your comment. We added a space before all brackets.

Round 2

Reviewer 2 Report

The manuscript has been extended and improved, but the language should be checked. Certain sentences are not readily clear and now and then prepositions are missed. Therefore, once the text is edited, the manuscript will be suitable for publication.